# Inhibition of *Salmonella* Enteritidis by Essential Oil Components and the Effect of Storage on the Quality of Chicken

**DOI:** 10.3390/foods12132560

**Published:** 2023-06-30

**Authors:** Wu Wang, Tingting Li, Jing Chen, Yingwang Ye

**Affiliations:** School of Food and Biological Engineering, Hefei University of Technology, Hefei 230009, China; 2021171473@mail.hfut.edu.cn (T.L.); chenjing_hfut@163.com (J.C.); yeyingwang04@126.com (Y.Y.)

**Keywords:** *Salmonella* Enteritidis, thymol, carvacrol, cinnamaldehyde

## Abstract

This research investigates the antibacterial potential of plant essential oil components including thymol, carvacrol, citral, cinnamaldehyde, limonene, and β-pinene against *Salmonella* Enteritidis (*S.* Enteritidis). Through the determination of minimum inhibitory concentration, three kinds of natural antibacterial agents with the best inhibitory effect on *S.* Enteritidis were determined, namely thymol (128 μg/mL), carvacrol (256 μg/mL), and cinnamaldehyde (128 μg/mL). Physical, chemical, microbial, and sensory characteristics were regularly monitored on days 0, 2, 4, and 6. The findings of this study reveal that both thymol at MIC of 128 μg/mL and carvacrol at MIC of 256 μg/mL not only maintained the sensory quality of chicken, but also decreased the pH, moisture content, and TVB-N value. Additionally, thymol, carvacrol and cinnamaldehyde successfully inhibited the formation of *S.* Enteritidis biofilm, thereby minimizing the number of *S.* Enteritidis and the total aerobic plate count in chicken. Hence, thymol, carvacrol, and cinnamaldehyde have more effective inhibitory activities against *S.* Enteritidis, which can effectively prevent the spoilage of chicken and reduce the loss of its functional components.

## 1. Introduction

In the course of the process of slaughter, circulation, sale, and processing, raw fresh meat is susceptible to contamination by *S.* Enteritidis, which can cause bacterial infections and even lead to food poisoning [1,2]. The control and elimination of *S.* Enteritidis in retail meat products poses a significant challenge. Tetracycline and sulfanilamide antibiotics are currently widely employed in poultry through drinking water and feed. However, many drug-resistant phenotypes of *Salmonella* isolates from food animals have been identified, and these are significantly higher than those found in human clinical isolates [3,4]. Consequently, the proficient surveillance of *S.* Enteritidis stands as an indispensable component in the deterrence and management of alimentary illnesses, thereby holding considerable import in the preservation of human well-being.

Plant essential oils (Eos) are considered viable alternatives to traditional antibacterial agents. The use of plant Eos as antibacterial agents not only implies greater safety for humans and a more environmentally friendly option due to their natural origin, but also represents a low risk of pathogenic microorganisms developing resistance. Thymol, carvacrol, and cinnamaldehyde have drawn considerable attention among researchers for their antibacterial potential [5,6]. Thymol, carvacrol, and cinnamaldehyde have broad-spectrum bacteriostatic effects. In the research of Qingliu Wei et al., a box-Behnken design model was selected, and 2.40 μL/mL cinnamon EO, 1.00 μL/mL cinnamon EO, and 3.50 μL/mL tea tree EO were optimized, which could effectively reduce the total aerobic plate count [7]. The antibacterial mechanism of cinnamon EO is achieved by exerting an impact on the respiratory metabolism, energy metabolism, and substance metabolism of *S.* Enteritidis, thereby resulting in a decrease in bacterial growth [8]. In the exploration of the impact of EOs on the adhesion of *Salmonella* strains and HT-29 cells, Alibi noted that six types of essential oils exhibited varying degrees of inhibitory effects on biofilm formation. Cinnamon oil demonstrated a remarkable inhibition rate of 99.10%, followed by clove oil with 97.64%, thyme oil with 95.90%, rosemary oil with 79.84%, turmeric oil with 28.98%, and sage oil with 15.55% [9]. The synergistic effect of different essential oils can change the permeability of the cell membrane, the leakage of dissolved substances in cells, the denaturation of membrane proteins, and the destruction of proton kinetics, thus achieving the antibacterial effect [10,11,12,13]. In addition, the use of natural preservatives and antimicrobial agents in polymer matrices has become a new trend. Peng et al. prepared antimicrobial packaging films from agar/konjac glucomannan (KA) and carvacrol (CV) and showed that the composite film containing 2% carvacrol was the most effective in preserving freshness and could extend the shelf life of frozen chicken breasts from 5 to 9 days [14]. Lian has arrived at the conclusion that thyme EO in chitosan composite film exhibits the highest release rate in a 50% ethanol solution. As the concentration of ethanol increases, the release rate of the essential oil from the composite film slows down, while the antibacterial activity of the film improves [15]. Meanwhile, researchers have been exploring nanoencapsulation technology to address the issues posed by lipid-soluble active ingredients that are insoluble in water or that are air-unstable or that have a pungent odor [16]. Although the incorporation of thymol crystals into zeolite networks has reduced the volatility of thymol, the strong aroma of the essential oil itself may alter the flavor of food, posing potential obstacles to certain applications [17]. Sodium alginate-based cinnamaldehyde controlled-release active packaging films loaded with halloysite nanotubes have broad prospects for application in high-fat food packaging [18]. In addition, the biodegradability of composite films, preparation costs, and potential risks associated with their use in food warrant further consideration and resolution by researchers [19].

Chicken is considered to be a major source of human *Salmonella* infection. Researchers have successfully isolated bacteriophages resistant to various *Salmonella* serotypes from raw chicken skin and claws [20]. Microbial infections not only result in the loss of nutrients and functional components in chicken meat, but also produce toxins that pose a threat to human health [21]. To effectively prevent and control *S.* Enteritidis in chicken meat, the use of safe and natural antimicrobial agents such as thymol, carvacrol, and cinnamaldehyde is of utmost importance [22]. Extensive research has been conducted on the antibacterial properties and inhibitory mechanisms of using plant essential oils as a method to suppress *Salmonella* in animal feed. However, there is limited research on the application of EOs for antibacterial and preservation purposes in fresh meat products [23,24,25]. The ultimate purpose of this study is to investigate the inhibitory effects of thymol, carvacrol, and cinnamaldehyde on *S.* Enteritidis, as well as their potential to improve the quality of chicken meat. Therefore, this research provides sufficient theoretical and practical basis for addressing the antibacterial and preservation issues of chicken.

## 2. Materials and Methods

### 2.1. Materials

Based on our preliminary research, an *S.* Enteritidis was screened from retail meat in the Hefei area [26]. Thymol (>99.0% purity), carvacrol (99% purity), limonene (>95.0% purity), citral (97% purity), cinnamaldehyde (≥95% purity), and β-pinene (purity ≥ 95%) were purchased from Shanghai Aladdin Biochemical Technology Co., Ltd. (Shanghai, China).

Buffered peptone water (BPW), bismuth sulfite agar (BSA), brilliant green sulfa agar (BGSA), Mueller-Hinton broth (MH), plate count agar (PCA), and tryptic soy broth (TSB) were purchased from Qingdao Haibo Biotechnology Co., Ltd. (Qingdao, China).

### 2.2. Minimum Inhibitory Concentration(MIC) Determinations

Initially, antimicrobial agents (such as thymol, carvacrol, limonene, citral, cinnamaldehyde, and β-pinene) at different concentrations (4096, 2048, 1024, 512, 256, 128, 64, 32, 16, 8, 4, 2, and1 μg/mL, respectively) were added into sterile 96-well polystyrene plates. Antimicrobial agents were added to wells 1–13, 10 μL per well, and well 14 was utilized as a growth control without the addition of antimicrobial agents. Subsequently, bacterial suspensions with turbidity equivalent to 0.5 McFarland standard were prepared using the direct bacterial suspension method and diluted 1:1000 in Mueller-Hinton broth. Then, 100 μL suspensions of *S.* Enteritidis were added to each well, each well was sealed, and each well was incubated for 48 h in ambient air at 37 °C to determine the results. The MIC is defined as the concentration where no visible bacterial growth is observed after 48 h of incubation at 37 °C under microaerophilic conditions [27].

### 2.3. Sample Preparation

The chicken used in this study was obtained from the Yong-hui Supermarket in Hefei. Three essential oil components with the smallest MIC were selected, and thymol at 128 μg/mL, carvacrol at 256 μg/mL, and cinnamaldehyde at 128 μg/mL were uniformly coated on fresh chicken, while the control group was coated with anhydrous ethanol. The samples were stored at 4 °C, and samples were taken on days 0, 2, 4, and 6, respectively. Under sterile conditions, four groups of samples (25–30 g each, treated with thymol, carvacrol, and cinnamaldehyde) were immersed in *S.* Enteritidis suspension with a concentration of 10^7^ CFU/g for 15 s, drained, and sampled on days 0, 2, 4, and 6.

### 2.4. Microbial Indicators Determinations

In the ultraclean workbench, 10 g of ground chicken was mixed with 9 mL of sterilized saline solution to create a 1:10 diluted sample. After shaking for 1 min, a series of 10-fold dilutions were performed and three appropriate dilutions were plated and incubated at 37 °C for 48 h to determine the total aerobic plate count [28].

Using the threefold dilution and most probable number (MPN) method, the amount of *S.* Enteritidis in chicken samples was determined. Specifically, 10 g of chicken meat was placed into a sterile homogenization bag containing 90 mL of BPW and homogenized for 1 min, and then 100 uL of the mixture was spread on BSA and BGSA plates, which were incubated at 37 °C for 24 h to analyze the presence of Salmonella colonies [29]. Following the procedure described in the Health Canada “Method Summary” (MFHPB-20), putative Salmonella colonies were isolated and subjected to biochemical and serological tests to confirm the identification of *S.* Enteritidis [30].

### 2.5. Antimicrobial Effects

A total of 20 μL of bacterial liquid (10^7^ CFU/mL) of *S.* Enteritidis isolates was added into a sterile 96-well plate, and 180 μL of thymol, carvacrol, and cinnamaldehyde were added to make their final concentrations 1/4 MIC, 1/2 MIC, and MIC respectively. Six parallel wells were provided for each strain, and 200 μL of TSB broth containing no bacteria was added to the negative control well, and then it was allowed to stand at 37 °C for 12 h. Crystal violet assay was performed to evaluate biofilm formation, and the optic density at 590 nm was measured. The average value is the result of at least three tests.

The ability of bacterial strain biofilm formation is classified into four categories [31]:

OD ≤ OD_C_ is a nonbiofilm former,

OD_C_ < OD ≤ 2OD_C_ is a weak biofilm former,

2OD_C_ < OD ≤ 4OD_C_ is a moderate biofilm former,

4OD_C_ < OD is a strong biofilm former.

Among them, the cutoff optical density value (OD_C_) is defined as three standard deviations higher than the average or average OD_AV_ of the negative control.

### 2.6. Physical and Chemical Indexes Determinations

#### 2.6.1. pH

By using a Five Easy Plus pH meter (Shanghai, China, Mettler Toledo Instrument Co., Ltd.), the electrode of the pH meter was directly placed in the meat tissue to determine the pH value [32]. The equipment was calibrated before measurement, and the experiment was repeated three times for each sample.

#### 2.6.2. Moisture Content

The moisture content of chicken was determined by the AOAC (2009) standard method.

#### 2.6.3. TVB-N

Referring to the method of Zhang et al., 10 g minced sample was added to 100 mL distilled water, homogenized for 1 min, and determined by automatic Kjeldahl nitrogen analyzer [33].

#### 2.6.4. Color

After the sample was ground, it was spread all over the bottom of the vessel, and immediately measured by a spectrophotometer. The spectrophotometer was preheated for 15 min before use and used after calibration, and recorded as brightness (L*), redness (a*), and yellowness (b*).

### 2.7. Sensory Evaluation

Six people with sensory evaluation experience formed the evaluation group, including three boys and three girls. Members of the evaluation team independently evaluated the color, smell, viscosity, and overall acceptability of each sample, and could not communicate with each other. All samples were graded according to the grading criteria in Table 1.

### 2.8. Statistical Analysis

The data were analyzed using one-way analysis of variance (ANOVA) and Duncan’s multiple range test in SPSS 14.0 software. Statistical significance was considered when *p* < 0.05. Additionally, data visualization graphs were created using Origin 2023 software.

## 3. Results

### 3.1. MIC

The compounds, namely thymol, carvacrol, limonene, citral, cinnamaldehyde, and β-pinene, exhibited a certain degree of inhibitory effect on *S.* Enteritidis. Table 2 displays the MIC values of six essential oil constituents against *S.* Enteritidis, with thymol, carvacrol, and cinnamaldehyde exhibiting MIC values of 128, 256, and 128 μg/mL, respectively, representing the three constituents with the smallest MIC values. Generally, a smaller MIC value indicates greater inhibition of bacterial growth, particularly in relation to *S.* Enteritidis. It is worth noting that EOs mainly composed of aldehydes and phenols have higher antibacterial activity, such as cinnamaldehyde, citral, carvacrol, and thymol, followed by EOs containing terpenoid alcohols [34]. Thymol and carvacrol have been found to disrupt the bacterial biofilm, causing the breakdown of the cell membrane and leakage of intracellular contents, ultimately resulting in cell death [12]. The aldehyde groups contained in citral and cinnamaldehyde can easily penetrate the cell wall and destroy the structure of polysaccharides. In addition, the aldehyde groups can also act on protein transporters, playing a role of sterilization [35,36]. Therefore, thymol, carvacrol, and cinnamaldehyde with low MIC were chosen for the next experiment.

### 3.2. Inhibitory Effect on S. Enteritidis Biofilm

From Figure 1, it can be observed that thymol, carvacrol, and cinnamaldehyde all have significant inhibitory effects on the formation of *S.* Enteritidis biofilms. As shown in Figure 1A, the OD value of the thymol-treated group at MIC significantly began to decrease compared to the control group and 1/4 MIC (*p* < 0.05) after 12 h of cultivation (early stage of biofilm formation). Therefore, it can be concluded that high concentrations of thymol have an inhibitory effect on *S.* Enteritidis biofilm formation at the early stage. At 24 h (basic biofilm formation) and 48 h (mature biofilm formation), the inhibitory effect of thymol on *S.* Enteritidis biofilms became more pronounced with increasing concentrations, as the OD values decreased with increasing concentrations, indicating that the ability of thymol to inhibit *S.* Enteritidis biofilms increased with increasing concentration. As shown in Figure 1B, carvacrol at MIC had a significant inhibitory effect on *S.* Enteritidis biofilm formation throughout the entire process (*p* < 0.05) compared to the control group. It is worth noting that the OD values of 1/2 MIC and 1/4 MIC were significantly higher than those of the control group at 12 h and 24 h, indicating that low concentrations of carvacrol had a promoting effect on *S.* Enteritidis biofilm formation at the early and basic formation stages. The OD values of 1/2 MIC and MIC were significantly lower than those of the control group at 12 h and 24 h, indicating that high concentrations of carvacrol had a strong inhibitory effect on *S.* Enteritidis biofilm formation at the early and mature stages. As shown in Figure 1C, the cinnamaldehyde-treated group had a significant inhibitory effect on *S.* Enteritidis biofilms at each concentration after 24 h and 48 h (*p* < 0.05). Using lower concentrations of EO will increase the permeability of the cell membrane and eventually lead to the destruction of cell membrane structures. Nevertheless, lower concentrations of EOs do not impact the viability of the cells. Conversely, the utilization of higher concentrations of EOs leads to extensive membrane damage and complete disruption of homeostasis, which ultimately results in the death of cells. Carvacrol and thymol can prevent or interfere with biofilm formation, and they can also eradicate preformed biofilms [37,38].

### 3.3. Changes in the Total Aerobic Plate Count of Chicken during Storage

During the storage of chicken, with the increase of days, the total aerobic plate count in each groups increased significantly (*p* < 0.05), as shown in Figure 2. Among them, the total aerobic plate count in the thymol treatment group and the carvacrol treatment group was significantly lower than those in the other two groups (*p* < 0.05), which indicated that thymol and carvacrol had obvious inhibitory effects on microbial reproduction in chicken during storage. However, the antibacterial effect of the cinnamaldehyde treatment group is not ideal. There was no significant difference between the two groups on the second and fourth days of storage (*p* > 0.05).

Similar to many other antibacterial agents and food preservation techniques, the effectiveness of cinnamaldehyde in food systems has been significantly reduced. This may be attributed to the inherent properties of chicken, such as ion strength and water activity, as well as differences in composition, including protein and fat content. These factors can affect the distribution of antibacterial agents within cells, thereby further diminishing their bactericidal activity [39]. On the fourth day of storage, the total aerobic plate count in the control group reached 6.22 Lg CFU/g, which exceeded the national fresh meat standard. Thymol, carvacrol, and cinnamaldehyde were all within the standard range of fresh meat, and total aerobic plate counts were 4.98 Lg CFU/g, 4.99 Lg CFU/g, and 5.98 Lg CFU/g, respectively. Meat is generally considered to be spoiled when the total aerobic plate count in meat products exceeds 6.0 Lg CFU/g [40]. On the sixth day of storage, all the treatment groups exceeded the fresh meat standard, indicating that the chicken in the control group could be stored for about two days, while thymol, carvacrol, and cinnamaldehyde had obvious antibacterial effects, which could prolong the storage time to about four days.

### 3.4. Changes in S. Enteritidis Count in Chicken Meat during Storage

In Figure 3, a comparison is made between the antibacterial effects of thymol, carvacrol, and cinnamaldehyde against *S.* Enteritidis during the storage of chicken. At day 0, the number of *S.* Enteritidis in each group was 4.39 Lg CFU/g, and the concentration of *S.* Enteritidis suspension used in the experiment was 7 Lg CFU/g. This indicates that a certain loss of *Salmonella* suspension occurred during the experimental operation, resulting in a final concentration of *S.* Enteritidis suspension dispersed on the surface of the chicken of 4.39 Lg CFU/g [41]. On the second day of storage, the number of *S.* Enteritidis in thymol and carvacrol treatment groups decreased to 4.20 Lg CFU/g and 3.96 Lg CFU/g, respectively, showing good antibacterial effect, which also proved that thymol and carvacrol had the strongest antibacterial effect in the first two days. However, the number of *S.* Enteritidis in the cinnamaldehyde treatment group and the control group had no significant difference in the first four days. Therefore, cinnamaldehyde’s ability to inhibit *S.* Enteritidis in chicken was weaker than thymol and carvacrol, which was confirmed in the research of Hoffman et al. [42] Furthermore, there are researchers who have proposed an order of antibacterial effectiveness of EOs as follows: thymol > oregano > carvacrol > trans-cinnamaldehyde > eugenol. Among these, thymol has demonstrated the strongest antibacterial effect against *Salmonella*, *Escherichia coli*, and *Klebsiella pneumoniae* [43]. With the extension of storage time, the number of *Salmonella* in each group increased rapidly. On one hand, due to the volatilization of thymol and carvacrol over time, the antibacterial effects of thymol and carvacrol are weakened. On the other hand, MIC thymol and carvacrol could not completely inhibit the growth of *S.* Enteritidis, but increased the lag time of *S.* Enteritidis. Although MIC concentrations of thymol and carvacrol in culture media were effective in inhibiting the growth of *S.* Enteritidis, the antimicrobial effect of applying MIC concentrations of thymol and carvacrol to chicken was diminished. Hence, the use of two or more complex EOs may need to be further considered in order to achieve similar antibacterial effects in food systems as in culture-based systems.

### 3.5. Changes in pH Values of Chicken during Storage

pH value is one of the most important factors affecting the quality of meat, which can affect the color, water retention capacity, flavor, tenderness, and shelf life of meat. A higher pH value is beneficial to the growth and reproduction of microorganisms, which means a short shelf life, while a lower pH value means a poor water retention ability [44]. The changes in the pH value of chicken during storage by thymol, carvacrol, and cinnamaldehyde are shown in Figure 4. The increase in microorganisms promotes the decomposition of protein in chicken meat, producing alkaline nitrogen-containing compounds such as amino acids, biogenic amines, and ammonia, which leads to aggravated meat spoilage and an increase in pH value. On the sixth day of storage, all groups experienced spoilage, indicating that chicken meat treated with thymol, carvacrol, and cinnamaldehyde had a shelf life of four days, delaying the spoilage of chicken meat. Compared with the control group, thymol, parsley phenol, and cinnamaldehyde can inhibit the growth and reproduction of microorganisms, reduce the pH value, and achieve a certain preservation effect.

### 3.6. Changes of Moisture Content of Chicken during Storage

The moisture content of fresh meat is an important factor affecting sensory quality and consumer perception [45]. During storage, the water content of chicken changes smoothly, as shown in Table 3. In general, the water content of chicken in each treatment groups did not change significantly during storage (*p* > 0.05). The decrease in moisture content of the control group during the first two days of storage can be attributed to the stiff state of the chicken meat, its poor water-holding capacity, and the loss of some free water during the sample preparation process. High moisture content in meat can lead to rapid putrefaction after death, as microorganisms proliferate rapidly in meat with a water content exceeding 80% [46]. From the second day of storage, the moisture contents of the thymol, carvacrol, and cinnamaldehyde treatment groups were higher than that of the control group, indicating that these three EOs inhibited the growth of microorganisms, slowed down the decomposition of chicken protein and fat, and thereby reduced the increase in moisture content.

### 3.7. Changes in the TVB-N of Chicken during Storage

The TVB-N value is an important quality indicator to evaluate the freshness of meat. The degradation of proteins and other nitrogenous compounds leads to the accumulation of organic amines, which are often referred to as total volatile basic nitrogen (TVB-N) [47]. The changes in the TVB-N content of chicken meat during storage for thymol, carvacrol, and cinnamaldehyde are shown in Figure 5, where TVB-N values in all treatment groups showed an increasing trend with time, similar to the changes in pH and total aerobic plate count. TVB-N values were as high as 20.17 mg/100 g in the control group and 18.13 mg/100 g in the cinnamaldehyde-treated group at day four of storage, both exceeding the fresh meat standard (≤15 mg/100 g), while thymol- and carvacrol-treated groups still did not exceed the standard, at 14.79 mg/100 g and 14.84 mg/100 g, respectively. The level of TVB-N in meat is related to the growth of microorganisms and the hydrolysis of protein into ammonia and amine. Chicken treated with thymol, carvacrol, and cinnamaldehyde leads to a decrease in the number of microorganisms responsible for protein degradation and the production of nonprotein nitrogen compounds including ammonia and amine. TVB-N significantly decreases compared to the control group. However, during storage, the antibacterial ability of thymol, carvacrol, and cinnamaldehyde gradually weakens, and the amplitude of TVB-N growth correspondingly increases. [48,49]. All treatment groups exceeded the standard on day six of storage, and the control group had a TVB-N value of 26.29 mg/100 g, which exceeded the standard for spoiled meat (>25 mg/100 g). The results indicate that thymol and carvacrol treatments can be effectively applied in chicken systems as natural bacterial inhibitors to extend the shelf life of chicken meat up to four days.

### 3.8. Changes in Chicken Color during Storage

The color of meat is an important indicator for consumers to evaluate the freshness of meat and make a purchase decision accordingly [50]. The changes in L*, a*, and b* values on the surface of chicken during storage are shown in Table 4. The L* values of each treatment group showed a trend of increasing first and then decreasing. The increase in L* values may be due to the destruction of the protein structure caused by the enzymatic degradation of protein, which leads to more light scattering [51]. The proliferation of microorganisms accelerates the spoilage and deterioration of chicken, causing a decrease in brightness, a sticky and darkened surface, and reduced elasticity after storage for four days. In the middle storage period, the brightness of the chicken treated with thymol and carvacrol was significantly higher than that of the control group (*p* < 0.05), which may be because thymol and carvacrol inhibited the growth of microorganisms and reduced the formation of methemoglobin, indicating that thymol and carvacrol treatment can improve the brightness of chicken. With the extension of storage time, the a* value of each treatment group showed a downward trend, while the b* values showed an upward trend. From the second day of storage, the red degrees of the thymol and carvacrol treatment groups were higher than that of control group (*p* < 0.05), which slowed down the oxidation of myoglobin to methemoglobin. On the sixth day, the red value decreased rapidly, which promoted the production of methemoglobin, so the change trend of the b* value was opposite to that of the a* value.

### 3.9. Changes in the Sensory Indicators of Chicken during Storage

Table 5 illustrates the sensory changes in chicken meat during storage when treated with thymol, carvacrol, and cinnamaldehyde. Within the first two days of storage, there were no significant differences in color, odor, or overall acceptability (*p* > 0.05) among the different treatment groups. However, there was a significant difference in viscosity between the thymol group and the control group (*p* < 0.05)*,* with only a slight stickiness observed on the surface of the chicken meat. On the fourth day of storage, there were significant differences in color, odor, viscosity, and overall acceptability (*p* < 0.05) between the thymol and carvacrol treatment groups and the control group, while only stickiness and overall acceptability (*p* < 0.05) showed significant differences between the cinnamaldehyde group and the control group, indicating a weaker preservation effect of cinnamaldehyde. The overall acceptability score of the control group was only 5.33, which was below the level that consumers would accept, with an odor score of 4, indicating a strong and unpleasant smell on the fourth day of storage, as well as a dull and sticky surface on the chicken meat. The overall acceptability score of the cinnamaldehyde group was also only 5.66, indicating a weak preservation effect, while the overall acceptability scores of the thymol and carvacrol treatment groups were both above 7, indicating that thymol and carvacrol could significantly inhibit the growth and reproduction of protein-degrading microorganisms. On the sixth day, the overall acceptability scores of all groups were around 3, indicating rapid microbial growth, which had penetrated into the chicken meat tissue and gradually decomposed to produce decay flavor substances such as dimethyl sulfide [49]. The antimicrobial properties of thymol, carvacrol, and cinnamaldehyde have been demonstrated to effectively minimize the microbial count, prolong the shelf life of products, and enhance the sensory quality of chicken.

## 4. Conclusions

In this paper, the inhibitory effects of thymol, carvacrol, and cinnamaldehyde on *S.* Enteritidis during storage and on the quality characteristics of chicken meat were evaluated. The results of physicochemical analysis showed that MIC (128 μg/mL) thymol- and MIC (256 μg/mL) carvacrol-treated chicken surfaces minimized the number of *S.* Enteritidis and total aerobic plate count, reduced pH and TVB-N, and extended the shelf life from two to four days compared to MIC (128 μg/mL) cinnamaldehyde. The results of sensory analysis showed that the color, odor, viscosity, and overall acceptability of the chicken remained good in the thymol and carvacrol groups. This study confirmed that thymol and carvacrol were more effective in inhibiting *S.* Enteritidis and improving chicken quality. Based on current research findings, our future efforts will focus on the development of an antibacterial film that combines thymol, carvacrol, cinnamaldehyde, and polysaccharides, which will be applied to fresh food packaging.

## Figures and Tables

**Figure 1 foods-12-02560-f001:**
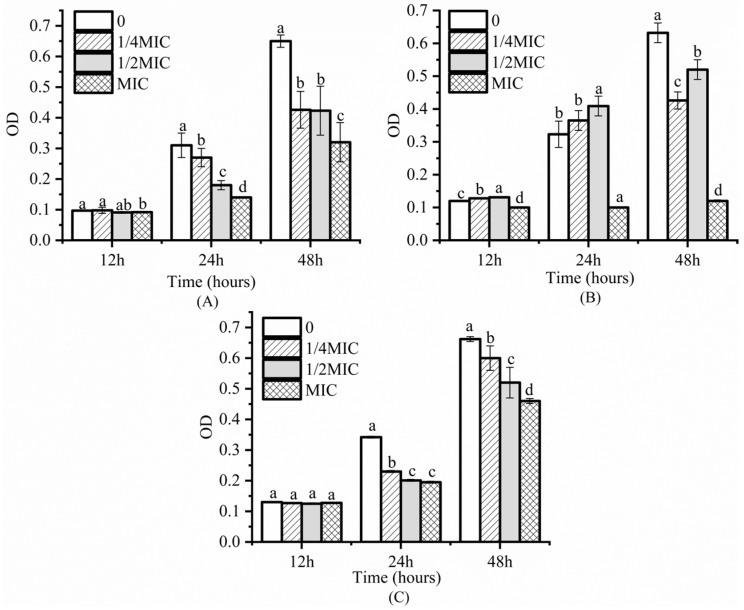
Inhibitory effects of thymol (**A**), carvacrol (**B**), and cinnamaldehyde (**C**) on biofilm formation in Salmonella. The different superscripts (a–d) indicate a statistically significant difference (*p* < 0.05) between different treatment groups within the same storage time.

**Figure 2 foods-12-02560-f002:**
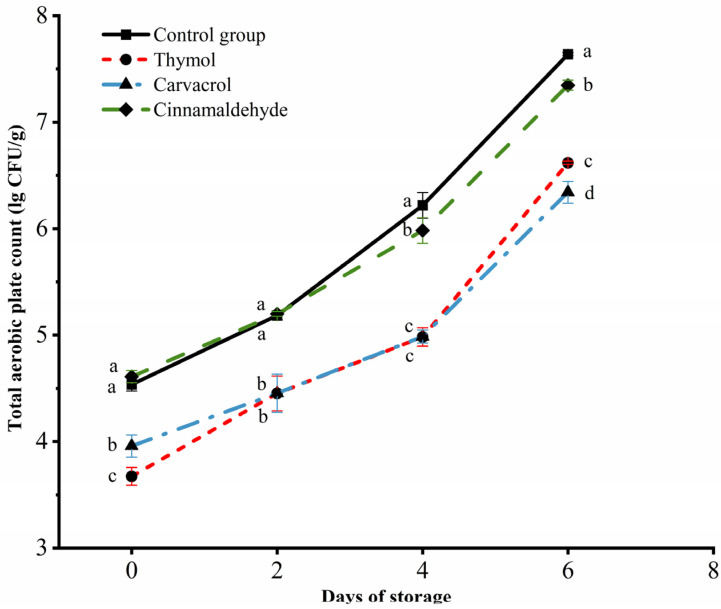
Effects of thymol, carvacrol, and cinnamaldehyde on total aerobic plate counts of chicken at storage. The different superscripts (a–d) indicate statistically significant differences (*p* < 0.05) between different treatment groups within the same storage time.

**Figure 3 foods-12-02560-f003:**
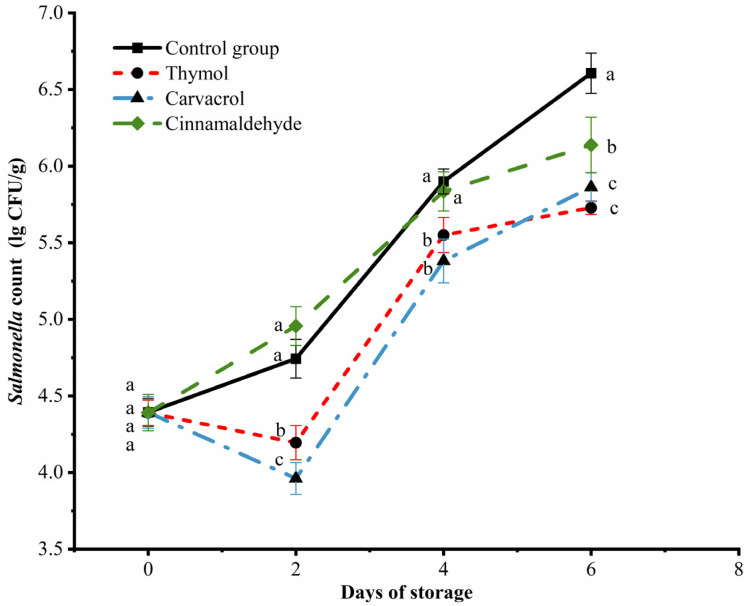
Effects of thymol, carvacrol, and cinnamaldehyde on *Salmonella* counts of chicken at storage. The different superscripts (a–c) indicate statistically significant differences (*p* < 0.05) between different treatment groups within the same storage time.

**Figure 4 foods-12-02560-f004:**
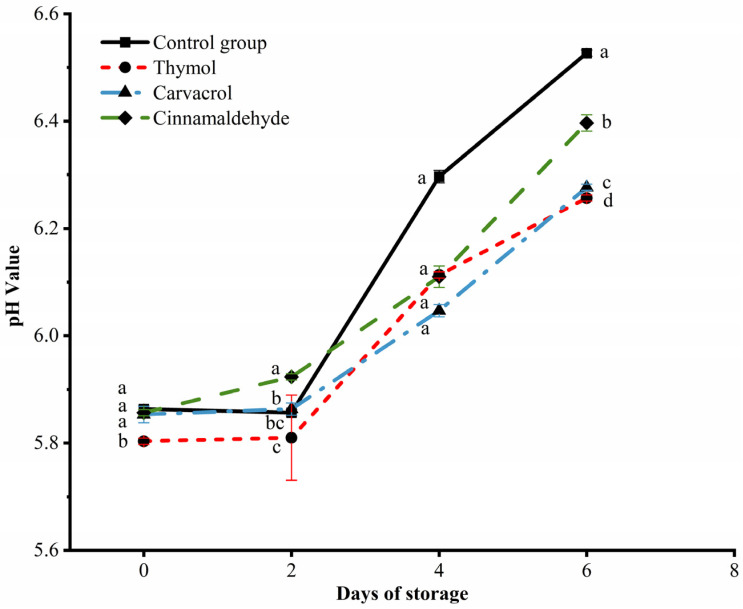
Effects of thymol, carvacrol, and cinnamaldehyde on the pH of chicken at storage. The different superscripts (a–d) indicate statistically significant differences (*p* < 0.05) between different treatment groups within the same storage time.

**Figure 5 foods-12-02560-f005:**
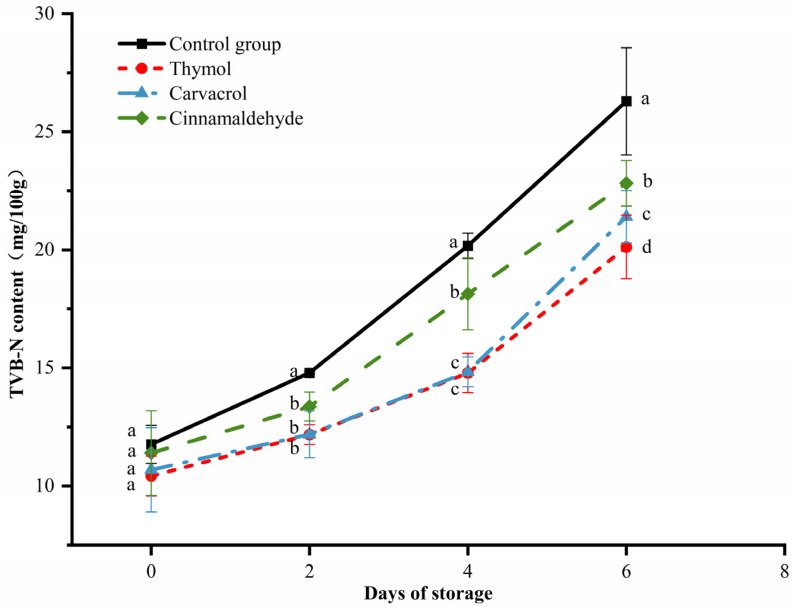
Effects of thymol, carvacrol, and cinnamaldehyde on the TVB-N of chicken at storage. The different superscripts (a–d) indicate statistically significant differences (*p* < 0.05) between different treatment groups within the same storage time.

**Table 1 foods-12-02560-t001:** Standards of sensory evaluation for chicken.

Indicators	Score	Scoring Criteria
Color	8–10 points	Light red, glossy surface
	6–7 points	Slightly dim, slightly shiny surface
	6 points or fewer	Dull, surface matte
Odor	8–10 points	Normal smell of chicken, no peculiar smell
	6–7 points	Slightly smelly
	6 points or fewer	Fishy odor or ammonia odor
Stickiness	8–10 points	Moist, not sticky
	6–7 points	Wetter, not sticky
	6 points or fewer	Sticky hands
Overall acceptability	8–10 points	Willing to accept
	6–7 points	Acceptable
	6 points or fewer	Unacceptable

**Table 2 foods-12-02560-t002:** Minimum inhibitory concentrations.

Essential Oil Main Ingredients	Thymol	Carvacrol	Limonene	Citral	Cinnamaldehyde	β-Pinene
MIC (μg/mL)	128	256	1024	512	128	2048

**Table 3 foods-12-02560-t003:** Effects of thymol, carvacrol, and cinnamaldehyde on the moisture content of chicken at storage.

Parameter	Processing Groups	Days of Storage			
		0 d	2 d	4 d	6 d
Moisture content	0	0.75 ^Aa^	0.74 ^Bb^	0.74 ^Bb^	0.74 ^Bb^
	Thymol	0.75 ^ABa^	0.75 ^Bab^	0.76 ^Aa^	0.75 ^ABa^
	Carvacrol	0.75 ^Aa^	0.75 ^Aab^	0.76 ^Aa^	0.76 ^Aa^
	Cinnamaldehyde	0.74 ^Ba^	0.75 ^ABa^	0.75 ^ABb^	0.76 ^Aa^

Different superscript letters (a,b) indicate statistically significant differences (*p* < 0.05) between different treatment groups for the same storage time; different superscript letters (A,B) indicate statistically significant differences (*p* < 0.05) between different storage times for the same treatment group.

**Table 4 foods-12-02560-t004:** Effects of thymol, carvacrol, and cinnamaldehyde on color of chicken at storage.

Parameter	Processing Groups	Days of Storage			
		0 d	2 d	4 d	6 d
L* values	0	54.87 ^Da^	56.37 ^Cb^	57.40 ^Bb^	55.27 ^Ab^
	Thymol	54.61 ^Da^	58.09 ^Ca^	58.42 ^Ba^	57.03 ^Aa^
	Carvacrol	54.86 ^Da^	57.93 ^Ca^	58.34 ^Ba^	56.21 ^Aab^
	Cinnamaldehyde	54.69 ^Da^	55.45 ^Cc^	57.36 ^Bb^	55.38 ^Abc^
a* values	0	5.13 ^Da^	4.55 ^Cb^	4.10 ^Bb^	3.49 ^Ab^
	Thymol	5.12 ^Da^	4.91 ^Ca^	4.78 ^Ba^	4.10 ^Aa^
	Carvacrol	5.07 ^Da^	4.99 ^Ca^	4.73 ^Ba^	3.74 ^Ab^
	Cinnamaldehyde	5.14 ^Da^	4.84 ^Ca^	4.13 ^Bb^	3.58 ^Ab^
b* values	0	31.54 ^Da^	32.71 ^Ca^	34.50 ^Ba^	35.93 ^Aa^
	Thymol	31.14 ^Dab^	32.12 ^Cab^	33.11 ^Bb^	34.60 ^Ac^
	Carvacrol	30.92 ^Db^	31.51 ^Cb^	32.31 ^Bc^	35.30 ^Ab^
	Cinnamaldehyde	31.29 ^Dab^	32.49 ^Ca^	34.34 ^Ba^	35.86 ^Aa^

Different superscript letters (a–c) indicate statistically significant differences (*p* < 0.05) between different treatment groups for the same storage time; different superscript letters (A–D) indicate statistically significant differences (*p* < 0.05) between different storage times for the same treatment group.

**Table 5 foods-12-02560-t005:** Effects of thymol, carvacrol, and cinnamaldehyde on the sensory parameters of chicken at storage.

Sensory Indicators	Processing Groups	0 d	2 d	4 d	6 d
Color	Control group	9.50 ± 0.84 ^Aa^	8.67 ± 1.21 ^Aa^	6.00 ± 0.89 ^Bb^	4.33 ± 0.82 ^Cb^
	Thymol	9.50 ± 0.84 ^Aa^	9.17 ± 0.98 ^Aa^	7.33 ± 0.82 _Ba_	6.17 ± 0.75 ^Ca^
	Carvacrol	9.33 ± 0.82 ^Aa^	9.17 ± 0.95 ^Aa^	7.83 ± 0.75 ^Ba^	5.83 ± 0.75 ^Ca^
	Cinnamaldehyde	9.33 ± 0.82 ^Aa^	8.83 ± 0.98 ^Aa^	6.83 ± 0.75 ^Bab^	4.16 ± 0.75 ^Cb^
Odor	Control group	9.67 ± 0.52 ^Aa^	8.67 ± 1.03 ^Aa^	4.00 ± 0.89 ^Bb^	2.33 ± 0.82 ^Cb^
	Thymol	9.83 ± 0.41 ^Aa^	9.00 ± 0.89 ^Aa^	6.50 ± 1.05 ^Ba^	3.17 ± 0.75 ^Cab^
	Carvacrol	9.67 ± 0.82 ^Aa^	9.17 ± 0.75 ^Aa^	5.67 ± 0.82 ^Ba^	3.50 ± 0.55 ^Ca^
	Cinnamaldehyde	9.50 ± 0.84 ^Aa^	8.83 ± 0.75 ^Aa^	3.83 ± 0.75 ^Bb^	2.50 ± 0.84 ^Cb^
Stickiness	Control group	9.83 ± 0.41 ^Aa^	8.33 ± 0.52 ^Bb^	6.83 ± 0.75 ^Cb^	5.83 ± 0.75 ^Da^
	Thymol	9.67 ± 0.52 ^Aa^	9.50 ± 0.84 ^Aa^	8.50 ± 1.05 ^Ba^	6.83 ± 0.75 ^Ca^
	Carvacrol	9.83 ± 0.41 ^Aa^	9.33 ± 1.03 ^ABab^	8.33 ± 1.37 ^Ba^	6.67 ± 1.03 ^Ca^
	Cinnamaldehyde	9.83 ± 0.41 ^Aa^	9.17 ± 0.98 ^ABab^	8.17 ± 0.98 ^Ba^	6.50 ± 0.84 ^Ca^
Overall acceptability	Control group	9.84 ± 0.41 ^Aa^	8.33 ± 1.03 ^Ba^	5.33 ± 0.82 ^Cb^	2.83 ± 0.75 ^Da^
	Thymol	9.67 ± 0.52 ^Aa^	9.33 ± 0.82 ^Aa^	7.17 ± 0.75 ^Ba^	3.50 ± 1.05 ^Ca^
	Carvacrol	9.67 ± 0.82 ^Aa^	9.00 ± 0.89 ^Aa^	7.00 ± 0.63 ^Ba^	3.50 ± 0.84 ^Ca^
	Cinnamaldehyde	9.67 ± 0.52 ^Aa^	9.00 ± 0.89 ^Aa^	5.67 ± 1.03 ^Bb^	3.00 ± 1.26 ^Ca^

Different superscript letters (a,b) indicate statistically significant differences (*p* < 0.05) between different treatment groups for the same storage time; different superscript letters (A–D) indicate statistically significant differences (*p* < 0.05) between different storage times for the same treatment group.

## Data Availability

The datasets generated for this study are available from the authors.

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
