# Peer review of "Inhibition of Salmonella Enteritidis by Essential Oil Components and the Effect of Storage on the Quality of Chicken"

_foods, 2023, doi:10.3390/foods12132560_

Round 1

Reviewer 1 Report

The graphs presented are difficult to understand. I could turn them all into tables to make it easier for readers who are not specific to the area to understand.

Author Response

We are grateful for the insightful suggestion of the reviewer, and we have addressed it here.

Q1.  The graphs presented are difficult to understand. I could turn them all into tables to make it easier for readers who are not specific to the area to understand.

Reply: Thank you for your comments. We have made adjustments to solve your problem about the difficulty in understanding charts. First, we added colors to represent different essential oil components and their changes in different storage periods. Secondly, we convert complex graphics into tabular format, which will help readers who are not familiar with this field to understand our results more easily.

In revised manuscript, Figures 1, 2, 3, 4 and 6 have been redrawn to clearly illustrate the changing trend of chicken quality in different storage periods. In addition, figs. 5 and 7 have been presented to readers in tabular form.

Reviewer 2 Report

Manuscript ID: foods-2454093

Title: Effects of thymol, carvacrol, and cinnamaldehyde on the quality of chicken during storage

Dear Editor

The manuscript is about the effect of three thymol, carvacrol, and cinnamaldehyde essential oil on S. Enteritidis isolated from chicken and its physicochemical and quality parameters.  Shelf life extension of food products, in particular, meat and chicken is important issue in food industry. Therefore, several studies have been focused on using natural components and essential oils in combination with other techniques such as edible coating/ films and packaging. It would be more interesting if that authors used EOs in coating or other treatments. Also, results were not discussing in depth and some predictable results are reported. This manuscript is needed major revision according to the comments:

·       Abstract: line 10 “S. Enteritidis”: All abbreviations in the text should be specified when they first appear. Scientific names should be Italic. Please check in all manuscript. Also. Please insert more details and results in abstract.

·       “2.2. Chicken and Sample Preparation” Section, line 84: How the MIC was measured? EOs are sensitive and volatile, so they degrade and lost their function. Thus, several methods are reported for using EOs in combination with edible coating and films or other packaging technique. Why didn’t use to improve the EOs efficacy?

·       What was the initial concentration of six EOs? And how minimum MIC was measured? please cited related reference for each method.’

·        What about the statistical analysis?? Please insert this section.

·       Figure 3: please use high quality and resolution of picture.

·       Line 223, Figure 3 >> 4

·       Line 237, Figure 4>> 5

·       Line 305, there are some topo?? (Odor and odor)

·       Why didn’t measured the antioxidant activity of EOs? DPPH or other?

·       Why didn’t measure PV and TBARS?

·       Please explain and discus the results in depth and use more recent refernces.

·       English should be improved.

-

Author Response

We appreciate that the reviewer recognizes the quality of our work. We are also grateful for the time and effort you have taken to provide insightful suggestion, and we have addressed it here.

Q1. Abstract: line 10 “S. Enteritidis”: All abbreviations in the text should be specified when they first appear. Scientific names should be Italic. Please check in all manuscript. Also. Please insert more details and results in abstract.

Reply: Thanks for your suggestion. The acronyms have been fixed, and the italicized passages have undergone an extensive examination and revision. The research objectives and research findings have been more explicitly stated in the revised abstract.

Q2. “2.2. Chicken and Sample Preparation” Section, line 84: How the MIC was measured? EOs are sensitive and volatile, so they degrade and lost their function. Thus, several methods are reported for using EOs in combination with edible coating and films or other packaging technique. Why didn’t use to improve the EOs efficacy?

Reply: We greatly appreciate this insightful question. Firstly, the specific operational steps for determining the MIC using the "Broth Microdilution Method" have been detailed in section 2.2. Secondly, with respect to the question of why the combination of coating, thin film, or other packaging technologies has not been utilized to enhance the efficacy of EOs, it is because this paper primarily investigates the antibacterial effects of plant essential oils against Salmonella and the preliminary results of the impact of added essential oil components on chicken meat quality. Therefore, the use of thin film and other packaging technologies to enhance the efficacy of essential oils has not yet been attempted. We appreciate your valuable feedback and will delve into this topic in future research.

Q3. What was the initial concentration of six EOs? And how minimum MIC was measured? please cited related reference for each method.

Reply: We express our deepest gratitude for this insightful inquiry. Six types of essential oils (EOs) were prepared at an initial concentration of 4096 μg/ml (using the double dilution method with subsequent dilutions of 4096, 2048, 1024, 512, 256, 128, 64, 32, 16, 8, 4, 2, 1 μg/mL). In addition, the minimum inhibitory concentration (MIC) was determined using the "Microbroth Dilution Method." Detailed operational procedures and references have been provided in the attachment for your perusal.

Q4. What about the statistical analysis? Please insert this section.

Reply: Thank you for your valuable suggestion. We did overlook statistical analysis, but we have now included that section in 2.8 as a supplement.

Q5. Figure 3: please use high quality and resolution of picture.

Reply: We greatly appreciate this insightful advice. In retrospect, we acknowledge our previous oversight and have since made necessary modifications to all images in order to enhance their quality and resolution. We apologize for any inconvenience that may have been caused during the course of your reading and extend our gratitude for your valuable suggestion.

Q6. Line 223, Figure 3 >> 4  Line 237, Figure 4>> 5

Reply: Thank you for your comments. We have made the necessary revisions to the numbering of all the figures and tables in the manuscript. We appreciate your prompt action in pointing out our mistake.

Q7. Line 305, there are some topo?(Odor and odor)

Reply: We greatly appreciate this insightful question. We have restructured the language and conveyed the research findings explicitly. We deeply apologize for any confusion or issues caused by inaccurate translation and express our gratitude for your valuable feedback.

Q7. Why didn’t measured the antioxidant activity of EOs? DPPH or other?

Why didn’t measure PV and TBARS?

Reply: We are very grateful for your thoughtful inquiry. We have indeed not tested the aforementioned indicators, primarily due to two reasons. Firstly, the storage time for chicken breast, according to Liu and Azimi's research, is rather short, and neither protein oxidation nor fat oxidation significantly alters throughout this time. Secondly, the paper focuses on the impact of antimicrobial agents on Salmonella and related quality indicators because there is little correlation between protein and fat oxidation and Salmonella. Our subsequent research will examine the impact of these antimicrobial agents on the degree of oxidation of chicken. We would like to express our gratitude for your inquiry once again.

Liu, G.; Song, H.; Zhang, Q.; Wang, J.; Wang, L.; Zhang, Z. Cellulose‐based Absorbent Pad Loaded with  Carum copticum  Essential Oil for Shelf Life Extension of Refrigerated Chicken Meat. Packag Technol Sci 2022, 35, 425–433, doi:10.1002/pts.2640.

Azimi, M.; Sharifan, A.; Ghiasi Tarzi, B. The Use of Pistacia Khinjuk Essential Oil to Modulate Shelf-Life and Organoleptic Traits of Mechanically Deboned Chicken Meat: PRESERVING MDPM PRODUCTS WITH PEO. Journal of Food Processing and Preservation 2017, 41, e12814, doi:10.1111/jfpp.12814.

Q8. Please explain and discuss the results in depth and use more recent references. English should be improved.

Reply: We appreciate your suggestions on our manuscript. We have made revisions to the results section by incorporating relevant literature to elucidate our research findings. Furthermore, we have endeavored to enhance our English writing skills. Once again, we express our gratitude for your valuable feedback which will assist us in improving the manuscript.

Reviewer 3 Report

Manuscript 2454093

Journal Foods

Title Effects of thymol, carvacrol, and cinnamaldehyde on the quality of chicken during storage

The manuscript entitled “Effects of thymol, carvacrol, and cinnamaldehyde on the quality of chicken during storage” describes the effect of thymol, carvacrol, and cinnamaldehyde against S. enteritidis ST2 on chicken with the evaluation of pH, moisture content, TVB-N, color, and sensory parameters during storage. The manuscript is interesting. However, the discussion of the results and several parts should be improved. Please follow the comments in the file.

English should be revised according to reviewer's comments.

Author Response

We appreciate that the reviewer recognizes the quality of our work. We are also grateful for the time and effort you have taken to provide insightful suggestion, and we have addressed it here.

Q1. L2-3 Please rewrite the title. The effect of essential oil compounds (and not only thymol, carvacrol, and cinnamaldehyde) on Salmonella (Salmonella is not mentioned) should be included.

Reply: We greatly appreciate your insightful comments. With reference to your comments, we have revised the title of the article to "Inhibition of Salmonella Enteritidis by essential oil components and the effect of storage quality of chicken ".

Q2. L9-19 Please rewrite the abstract. Quantitative data should be included.

Reply: We thank the reviewer’s useful suggestion. We have meticulously revised the abstract content and included specific data to present our research findings. Please refer to the attached document for further details.

Q3. L9 Here and throughout the manuscript please use italics for bacterial species.

Reply: Thanks for your valuable suggestion. We must acknowledge that we have neglected this issue, and have now adjusted all references to bacterial species in the text to italicize them. We are grateful for having this matter brought to our attention again.

Q4. L34-37 Please check this part. The reference 4 is not Qingliu Wei et al.

Reply: We grateful for the insightful suggestion of the reviewer. This is caused by our careless inspection. We have corrected it and checked all the references one by one.

Q5. L37-40 Please revise this part. Better antimicrobial effect?

Reply: Thanks for your useful suggestion.  Examples show that thymol, carvacrol and cinnamaldehyde have the potential mechanism of broad-spectrum antibacterial action, that is, they may change the permeability of bacterial cell membrane, increasing the leakage of intracellular dissolved substances, denature membrane proteins, destroy proton kinetics, and thus inhibit the growth of pathogenic bacteria.

Q6. L40-44 Please replace these examples with those related to the antibacterial action of thymol, carvacrol, and cinnamaldehyde against S. Enteritidis.

Reply: We are grateful for your insightful suggestion. We have replaced the original content with reference materials that are more closely associated with the subject matter of this manuscript.

Zhang, Z.; Zhao, Y.; Chen, X.; Li, W.; Wang, L.; Li, W.; Du, J.; Zhang, S. Effects of Cinnamon Essential Oil on the Physiological Metabolism of Salmonella Enteritidis. Front. Microbiol. 2022, 13, 1035894, doi:10.3389/fmicb.2022.1035894.

Alibi, S.; Selma, W.B.; Mansour, H.B.; Navas, J. Activity of Essential Oils Against Multidrug-Resistant Salmonella Enteritidis. Curr Microbiol 2022, 79, 273, doi:10.1007/s00284-022-02938-x.

Q7. L48-50 Please be more specific. Add details on the essential oil and other information.

Reply: We are grateful for your insightful suggestion. We have provided additional clarification on the content of the cited references, and we express our gratitude for bringing to our attention the inadequacies in our work.

Q8. L50-56 Please expand this part including the use of different carriers to protect essential oil and drive their release in food matrices. The papers doi.org/10.3390/ijms23169227, doi.org/10.1016/j.heliyon.2022.e09551, and doi.org/10.3390/foods10061150 are suggested for your analysis and discussion.

Reply: Thanks for your useful comments. We meticulously studied the three references you provided, and deeply appreciated them after removing extraneous content from the original references. We conducted a thorough analysis and discussion of the literature, and synthesized our conclusions in the manuscript. We extend our sincere appreciation for your attentive guidance.

Q9. L62-64 Please be more specific. Better describe the results of these references.

Reply: Thanks for your useful comments. The original content has been modified as a result of a deeper understanding of the application of plant essential oils in inhibiting Salmonella Enteritidis. Previous research has focused primarily on the antibacterial activity and inhibition mechanisms of essential oils, rather than their potential to inhibit Salmonella Enteritidis during the storage of fresh chicken meat and maintain its quality. This lack of research has prompted the focus of this study.

Q10. L64-68 Please rewrite the objective including all the essential oil compounds considered and all the experimental steps.

Reply: Thanks for your useful comments. We have revised the research objectives and clarified the significance of the study. We extend our gratitude for the valuable insights you have provided.

Q11. Please divide into 2 sections: the first one related to the MIC determination, the second one related to challenge test on chicken.

Reply: Thanks for your useful comments. We have already disaggregated this section of the content, providing a detailed exposition of the procedural steps. For further information, please refer to the attached document.

Q12. L80-83 Please better describe the MIC determination including the preparation of the inoculum, the concentration assayed, and the interpretation of the results.

Reply: Thanks for your useful comments. The specific operational steps for determining the MIC using the "Broth Microdilution Method" have been detailed in section 2.2. For further information, please refer to the attached document.

Q13. L90-96 Rewrite. It is not correct in English. Moreover, more details should be added. How Salmonella was determined? Please be more specific.

Reply: We thanks for your useful suggestion. The specific procedures for determining the total number of bacterial colonies and the quantity of Salmonella have been documented in section 2.4. For further information, please refer to the attached document.

Q14. L104-111 It is not clear. Please better explain in the text.

Reply: Thanks for your useful suggestion. The optical density at 590 nm is the OD value, and the average OD value of the negative control plus its standard deviation SD is defined as the critical value ODc. The biofilm-forming ability of the strains can be divided into four types: OD ≤ ODc is a non-film-forming strain, ODc < OD ≤ 2ODc is a weak film-forming strain, 2ODc < OD ≤ 4ODc is a medium film-forming strain, and OD> 4ODc is a strong film-forming strain. We are uncertain whether our response has met your satisfaction. We extend our gratitude for your inquiry.

Q15. L118-123 How these parameters were expressed in tables/figures? Please add this information.

Reply: We thanks for your useful suggestion. The description of the methods chosen for determining moisture content and total volatile basic nitrogen are presented in lines 118-123, accompanied by detailed results in sections 3.6 and 3.7.

Q16. L130 Why only 6 panelists were considered? Please explain 

Reply: We are grateful for your insightful suggestion.   We have solicited the expertise of six doctoral individuals with extensive experience in flavor research to conduct sensory evaluations within the meat products team of the School of Food and Biological Engineering at Hefei University of Technology. Their rigorous and standardized judgement criteria have enhanced the relevance of the scoring outcomes.

Q17. L138-139 Add the statistical analysis section.

Reply: We express our gratitude to you for providing a valuable suggestion.  We did overlook statistical analysis, but we have now included that section in 2.8 as a supplement.

Q18. L141 Delete and so on.

Reply: We are grateful for your insightful suggestion. The term "and so on" has been successfully removed from the text. We express our gratitude for bringing our attention to this error in a timely manner.

Q19. L141-144 Rewrite. It is not correct in English. Rephrase.

Reply: We express our gratitude for your valuable suggestion. We have re-written the aforementioned section to address the issue of sentence disorder. We apologize for any inconvenience this may have caused during your review and kindly ask for your understanding.

Q20. L154 Table 2 I see a MIC of 128 ug/mL for citral. Why this compound was not selected? Please explain in the manuscript and in this section.

Reply: Thanks for your insightful recommendations.We thought about your problem seriously, and finally found that it was caused by data writing errors. We acknowledge that during the preparation of the submission, we did not thoroughly examine the accuracy of the data, resulting in unnecessary complications and delays to your work. Please accept our sincere apologies for any inconvenience caused. We assure you that we will learn from this experience and exercise greater caution in our future work to ensure that similar errors do not occur again. Once again, we apologize and hope that you can accept our heartfelt apology and grant us the opportunity to demonstrate our abilities.

Q21. L156-176 Replace total number of colonies with total aerobic plate count.

Reply: Thank you for your comments. We have incorporated their suggestions and made corrections to all areas of the article where the total aerobic plate count was reported. This experience highlights the importance of maintaining accuracy and accuracy when using technical terms in the writing process. Once again, we extend our appreciation for the valuable insights provided.

Q22. L176 Replace 4 days with 2 days.

Reply: We express our gratitude to the reviewer for your insightful recommendations. As stated by Zhao et al, meat is generally considered to be spoiled when the total aerobic plate count in meat products exceeded 6.0 Lg CFU/g. On the fourth day of storage, the total aerobic plate count in the control group reached 6.22 Lg CFU/g, which exceeded the national fresh meat standard. Thymol, carvacrol and cinnamaldehyde were all within the standard range of fresh meat, and total aerobic plate count being 4.98 Lg CFU/g, 4.99 Lg CFU/g and 5.98 Lg CFU/g respectively. Therefore, we believe that the wording in the text is correct. We have revised the sentence because we may not have expressed it accurately enough and caused misunderstanding to your reading. Should our interpretation prove to be misguided, we welcome further communication in order to rectify any errors. We extend our gratitude for your invaluable input.

Zhao, R.; Zhang, Y.; Chen, H.; Song, R.; Li, Y. Performance of Eugenol Emulsion/Chitosan Edible Coating and Application in Fresh Meat Preservation. Food Processing Preservation 2022, 46, doi:10.1111/jfpp.16407.

Q23. L178-179 Here and in all figures/tables add a footnote describing the statistical analysis applied and the meaning of different letters.

Reply: We express our gratitude for the valuable suggestion provided by the reviewer. We have appended explanatory footnotes to each of the charts and tables. Additionally, we have provided a detailed account of the statistical analysis employed and elaborated on the connotations of distinctive symbols.

Q24. L180-196 Please include data related to cinnamaldehyde and better discuss the results with data from literature. Please improve the discussion of the results

Reply: We express our gratitude for your valuable suggestion. We supplemented the antibacterial effect of cinnamaldehyde in the figure, and our research results are similar to those of Liu et al.

Hoffman-Pennesi, D.; Wu, C. The Effect of Thymol and Thyme Oil Feed Supplementation on Growth Performance, Serum Antioxidant Levels, and Cecal Salmonella Population in Broilers. Journal of Applied Poultry Research 2010, 19, 432–443, doi:10.3382/japr.2009-00141.

Liu, X.; Liu, R.; Zhao, R.; Wang, J.; Cheng, Y.; Liu, Q.; Wang, Y.; Yang, S. Synergistic Interaction Between Paired Combinations of Natural Antimicrobials Against Poultry-Borne Pathogens. Front. Microbiol. 2022, 13, 811784, doi:10.3389/fmicb.2022.811784.

Q25. L193-196 Why authors did not use a concentration higher than MIC values in this challenge-test? Please explain.

Reply: We express our gratitude for your valuable suggestion.As Liu et al. suggested thymol, carvacrol and cinnamaldehyde have an extremely strong bacteriostatic effect and inhibit the growth of undesirable microorganisms in food. Although the antibacterial effect at higher concentrations is obvious, the addition of higher concentrations of thymol, carvacrol and cinnamaldehyde produces a strong flavor (as also pointed out by Cometa) that can mask the flavor of the chicken and thus negatively affect the sensory characteristics of the chicken. Therefore, we did not choose concentrations higher than the MIC values in our experiments. We extend our gratitude for your invaluable input.

Liu, X.; Liu, R.; Zhao, R.; Wang, J.; Cheng, Y.; Liu, Q.; Wang, Y.; Yang, S. Synergistic Interaction Between Paired Combinations of Natural Antimicrobials Against Poultry-Borne Pathogens. Front. Microbiol. 2022, 13, 811784, doi:10.3389/fmicb.2022.811784.

Cometa, S.; Bonifacio, M.A.; Bellissimo, A.; Pinto, L.; Petrella, A.; De Vietro, N.; Iannaccone, G.; Baruzzi, F.; De Giglio, E. A Green Approach to Develop Zeolite-Thymol Antimicrobial Composites: Analytical Characterization and Antimicrobial Activity Evaluation. Heliyon 2022, 8, e09551, doi:10.1016/j.heliyon.2022.e09551.

Q26. L200 Please add Salmonella biofilm.

Reply: Thanks for your valuable suggestion. Referring to your suggestion, we have made modifications to the title of section 3.2. Thank you again for your precise feedback.

Q27. L210-214 Replace with studies on the antibiofilm activity of these compounds against Salmonella. 

Reply: We express our gratitude for the valuable suggestion provided by the reviewer. We have supplemented relevant literature on the effects of essential oils on Salmonella biofilms and provided explanations in the text. Please refer to the attachment.

Q28. L238-242 Please rewrite. It is not clear.

Reply: We express our gratitude for the valuable suggestion provided by the reviewer. We have rephrased the aforementioned portion of text. We sincerely apologize for any inconvenience this may have caused during your reading. We would like to extend our gratitude for your valuable feedback.

Q29. L252 Replace volatile salt nitrogen values with total volatile basic nitrogen values.

Reply: We express our gratitude to the esteemed reviewer for providing a valuable suggestion. We adopted your suggestion and corrected all the references to total volatile basic nitrogen in the article. This experience highlights the importance of maintaining accuracy and accuracy when using technical terms in the writing process. Once again, we are grateful for the valuable insights provided.

Q30. L252-269 Please add a discussion of these results including data from literature. The papers doi.org/10.1002/pts.2640, doi.org/10.1016/j.foodhyd.2020.106011, doi.org/10.1016/j.fpsl.2021.100751 are suggested for your analysis and discussion.

Reply: We thank the reviewer’s useful suggestion. We meticulously studied the three references you provided, and deeply appreciated them after removing extraneous content from the original references. We conducted a thorough analysis and discussion of the literature, and synthesized our conclusions in the manuscript. We extend our sincere appreciation for your attentive guidance.

Q31. L280 rotted? Please check this term. It is not correct in English.

Reply: We grateful for your  insightful suggestion. We found "rotted" and corrected the entire sentence. Thank you for promptly pointing out our errors.

Q32. L273-292 Why authors did not determine myoglobin content? Please explain.

Reply: We are deeply impressed by your question. The theme of the manuscript is the inhibitory effect of thymol, carvacrol and cinnamaldehyde on Salmonella and the influence of chicken storage quality. This section focuses on the change of chicken color during storage. At present, it has been concluded that the a* value of all treatment groups decreases with the extension of storage time, on the contrary, the b* value shows an upward trend, and it is explained. We apologize for not conducting further investigations into the mechanism. But we will discuss the changes of myoglobin and methemoglobin content during storage later. We grateful for the insightful suggestion of the reviewer.

Q33. L295-311 Please better describe the results of Table 3 and include a discussion of these results.

Reply: We thank the reviewer’s useful suggestion.  We have analyzed the scores of the sensory evaluation scale and explained the reasons for them. Thank you for your suggestions that enhance the academic value of our findings.

Q34. L317-327 Rewrite the conclusion section. Please include data on the biofilm formation of Salmonella treated with different compounds. 

Reply: Thanks for your useful comments.  We have rephrased the conclusion to reflect the research topic and expound upon the findings. Please refer to the attached document for further details.

Q35. L324-327 Rewrite this part. Authors have no data on the decomposition of protein and fat in chicken, and the formation of methemoglobin. Delete or rewrite.

Reply: We thanks for your useful suggestion. The present text indeed lacks data on the breakdown of protein and fat in chicken, as well as the formation of methemoglobin. Therefore, we have deleted this section and rewritten the conclusion, clarifying the research topic and elaborating on the findings. We extend our sincere appreciation for your attentive guidance.

Q36. Why authors did not determine the biofilm formation on chicken surface? Please explain.

Reply: Thank you for your valuable advice. My view of the issue is that, in the first place, a biofilm is a collection of cells that are bound together by extracellular polymers generated by the bacteria themselves. This enables the bacteria to aggregate and adhere and, as a result, resist the harmful external environment. The effect of thymol, carvacrol, and cinnamaldehyde on the Salmonella enterica biofilm is discussed in this paper. As the concentration of essential oils rises, this will eventually affect Salmonella enterica growth and reproduction, causing damage to the biofilm and resulting in the effect of bacterial inhibition. Second, the goal of our study was to examine how Salmonella enterica was inhibited by thymol, carvacrol, and cinnamaldehyde as well as how well chicken flesh stored, so we did not consider the determination of biofilm formation on the surface of chickens. We extend our gratitude for your invaluable input.

Q37. L346-438 Revise the reference list including other references and revise the numbering of references in the text.

Reply: Thank you very much for pointing out our problems in time. We check the contents and serial numbers of the references one by one to ensure that they are correct and we apologize for the inconvenience caused to your reading.

Round 2

Author Response

We appreciate that the reviewer recognizes the quality of our work. We are also grateful for the time and effort you have taken to provide insightful suggestion, and we have addressed it here.

Q1. Usually the microbial and physicochemical quality (PV and TBARS) of chicken or other meat products during the storage are investigated. If the main goal of this study was the anti-bacterial effect of EOs against S. Enteritidis, so should be appear in the title. Therefore, I suggest authors modify title to “Inhibition of Salmonella Enteritidis by Essential Oil Components and the Their Effect on Microbial Quality of Chicken During the Storage”.

Reply: Thank you for your comments. Initially, based on the research conducted by Liu and Azimi, it was found that the storage duration of chicken is relatively short and during this period, there is no significant change in the oxidation of protein and fat. Therefore, PV and TBARS were not measured. Secondly, the purpose of this paper is to study the inhibitory effects of thymol, carvacrol and cinnamaldehyde on Salmonella enteritidis and their potential to maintain chicken quality. Hence, we have considered your proposal and changed the title to " Inhibition of Salmonella Enteritidis by Essential Oil Components and the Effect of Storage on the Quality of Chicken". We extend our sincere appreciation for your attentive guidance.

Liu, G.; Song, H.; Zhang, Q.; Wang, J.; Wang, L.; Zhang, Z. Cellulose‐based Absorbent Pad Loaded with  Carum copticum  Essential Oil for Shelf Life Extension of Refrigerated Chicken Meat. Packag Technol Sci 2022, 35, 425–433, doi:10.1002/pts.2640.

Azimi, M.; Sharifan, A.; Ghiasi Tarzi, B. The Use of Pistacia Khinjuk Essential Oil to Modulate Shelf-Life and Organoleptic Traits of Mechanically Deboned Chicken Meat: PRESERVING MDPM PRODUCTS WITH PEO. Journal of Food Processing and Preservation 2017, 41, e12814, doi:10.1111/jfpp.12814.

Q2. Line 87; Salmonella Enteritidis. Also, please use the abbreviation (S. Enteritidis) after first appear.

Reply: Thanks for your suggestion. We have corrected the abbreviation of Salmonella Enteritidis in the text, thank you again for bringing this to our attention.

Q3. Please complete the conclusion and suggest the combination of edible coating/film or packaging with these EOs for chicken preservation as future trends.

Reply: We greatly appreciate this insightful advice. In the concluding section, an additional avenue for future research is proposed, which involves the incorporation of edible coatings/films or packaging with thymol, carvacrol, and cinnamaldehyde for the preservation of chicken.

Q4. English should be improved.

Reply: Thanks for your suggestion. We have made great efforts in reading English literature and practicing English writing, in order to enhance our proficiency in the language. We highly appreciate and value the valuable recommendations and suggestions you have provided us.

Reviewer 3 Report

Authors addressed large part of the reviewer's comments. Some minor changes are suggested below:

L5-7 ...and the effect of storage on the quality of chicken

L35 Please add a conclusion in the abstract

L84 Here and throughout the manuscript please use only total aerobic plate count and not total aerobic plate counts of colonies

L209-215 Delete. It is not necessary

L413-415 Please rewrite. It is not clear

L518-522 Please better explain. It is not clear

L539-541 Rewrite. It is not correct and not clear. Please revise this sentence

Some sentences could be improved. Please follow the comments

Author Response

We appreciate that the reviewer recognizes the quality of our work. We are also grateful for the time and effort you have taken to provide insightful suggestion, and we have addressed it here.

Q1. L5-7 ...and the effect of storage on the quality of chicken.

Reply: Thanks for your suggestion. After taking your advice into consideration, we have made modifications to the title. Please refer to the attached document for more information.

Q2. L35 Please add a conclusion in the abstract. 

Reply: Thank you for your comments. We have added concluding statements to the summary section, and we express our gratitude for your valuable suggestions.

Q3. L84 Here and throughout the manuscript please use only total aerobic plate count and not total aerobic plate counts of colonies.

Reply: We express our deepest gratitude for this insightful suggestion. We carefully checked all the places where the total aerobic plate count appeared in the entire manuscript and corrected some of the writing errors.

Q4. L209-215 Delete. It is not necessary.

Reply: Thank you for your valuable suggestion. We have deleted the unnecessary contents such as the preparation of culture medium and the operation process of normal saline and we thank you for promptly pointing out our errors.

Q5. L413-415 Please rewrite. It is not clear.

Reply: We greatly appreciate this insightful advice. To facilitate the reader to better understand the study results, we have provided additional explanations for this section. Thank you again for your proposal.

Q6. L518-522 Please better explain. It is not clear.

Reply: Thank you for your comments. We have further explained the content of this section apologize for any inconvenience caused to your reading. Please refer to the attachment for more information. We extend our gratitude for your suggestion.

Q7. L539-541 Rewrite. It is not correct and not clear. Please revise this sentence.

Reply: We greatly appreciate this insightful advice. We have reanalyzed and explained this part and we thank you for promptly pointing out our errors.
